# LATENT SPACE ODDITY: ON THE CURVATURE OF DEEP GENERATIVE MODELS

**Georgios Arvanitidis, Lars Kai Hansen, Søren Hauberg**
Technical University of Denmark, Section for Cognitive Systems
`{gear,lkai,sohau}@dtu.dk`

## ABSTRACT

Deep generative models provide a systematic way to learn nonlinear data distributions through a set of latent variables and a nonlinear "generator" function that maps latent points into the input space. The nonlinearity of the generator implies that the latent space gives a distorted view of the input space. Under mild conditions, we show that this distortion can be characterized by a stochastic Riemannian metric, and we demonstrate that distances and interpolants are significantly improved under this metric. This in turn improves probability distributions, sampling algorithms and clustering in the latent space. Our geometric analysis further reveals that current generators provide poor variance estimates and we propose a new generator architecture with vastly improved variance estimates. Results are demonstrated on convolutional and fully connected variational autoencoders, but the formalism easily generalizes to other deep generative models.

## 1 INTRODUCTION

Deep generative models (Goodfellow et al., 2014; Kingma & Welling, 2014; Rezende et al., 2014) model the data distribution of observations $\mathbf{x} \in \mathcal{X}$ through corresponding latent variables $\mathbf{z} \in \mathcal{Z}$ and a stochastic *generator function* $f : \mathcal{Z} \to \mathcal{X}$ as

$$\mathbf{x} = f(\mathbf{z}). \tag{1}$$

Using reasonably low-dimensional latent variables and highly flexible generator functions allows these models to efficiently represent a useful distribution over the underlying data manifold. These approaches have recently attracted a lot of attention, as deep neural networks are suitable generators which lead to the impressive performance of current *variational autoencoders (VAEs)* (Kingma & Welling, 2014) and *generative adversarial networks (GANs)* (Goodfellow et al., 2014).

Consider the left panel of Fig. 1, which shows the latent representations of digits 0 and 1 from MNIST under a VAE. Three latent points are highlighted: one point (A) far away from the class boundary, and two points (B, C) near the boundary, but on opposite sides. Points B and C near the boundary seem to be very close to each other, while the third is far away from the others. Intuitively, we would hope that points from the same class (A and B) are closer to each other than to members of other classes (C), but this is seemingly not the case. *In this paper, we argue this seemed conclusion is incorrect and only due to a misinterpretation of the latent space — in fact points A and B are closer to each other than to C in the latent representation.* Correcting this misinterpretation not only improves our understanding of generative models, but also improves interpolations, clusterings, latent probability distributions, sampling algorithms, interpretability and more.

In general, latent space distances lack physical units (making them difficult to interpret) and are sensitive to specifics of the underlying neural nets. It is therefore more robust to consider infinitesimal distances along the data manifold in the input space. Let $\mathbf{z}$ be a latent point and let $\Delta \mathbf{z}_1$ and $\Delta \mathbf{z}_2$ be infinitesimals, then we can compute the squared distance

$$\|f(\mathbf{z} + \Delta \mathbf{z}_1) - f(\mathbf{z} + \Delta \mathbf{z}_2)\|_2^2 = (\Delta \mathbf{z}_1 - \Delta \mathbf{z}_2)^\intercal (\mathbf{J}_\mathbf{z}^\intercal \mathbf{J}_\mathbf{z}) (\Delta \mathbf{z}_1 - \Delta \mathbf{z}_2), \quad \mathbf{J}_\mathbf{z} = \left.\frac{\partial f}{\partial \mathbf{z}}\right|_{\mathbf{z}=\mathbf{z}}, \quad (2)$$

using Taylor's Theorem. This implies that the natural distance function in $\mathcal{Z}$ changes locally as it is governed by the local Jacobian. Mathematically, the latent space should not then be seen

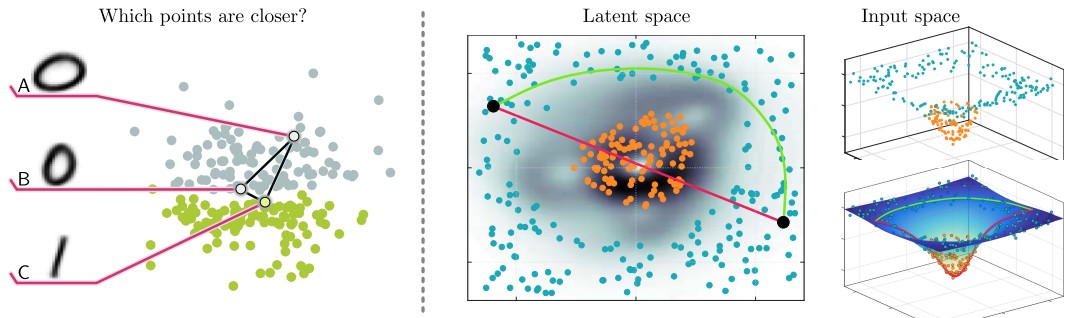

Figure 1: *Left:* An example of how latent space distances do not reflect actual data distances. *Right:* Shortest paths on the surface spanned by the generator do not correspond to straight lines in the latent space, as is assumed by the Euclidean metric.

as a linear Euclidean space, but rather as a curved space. The right panel of Fig. 1 provides an example of the implications of this curvature. The figure shows synthetic data from two classes, and the corresponding latent representation of the data. The background color of the latent space corresponds to $\sqrt{\det(\mathbf{J_z^\intercal J_z})}$, which can be seen as a measure of the local distortion of the latent space. We interpolate two points from the same class by walking along the connecting straight line (red); in the right panel, we show points along this straight line which have been mapped by the generator to the input space. Since the generator defines a surface in the input space, we can alternatively seek the shortest curve along this surface that connects the two points; this is perhaps the most natural choice of interpolant. We show this shortest curve in green. From the center panel it is evident that the natural interpolant is rather different from the straight line. This is due to the distortion of the latent space, which is the topic of the present paper.

**Outline.** In Sec. 2 we briefly present the VAE as a representative instance of generative models. In Sec. 3 we connect generative models with their underlying geometry, and in Sec. 4 we argue that a stochastic Riemannian metric is naturally induced in the latent space by the generator. This metric enables us to compute length-minimizing curves and corresponding distances. This analysis, however, reveals that the traditional variance approximations in VAEs are rather poor and misleading; we propose a solution in Sec. 4.1. In Sec. 5 we demonstrate how the resulting view of the latent space improves latent interpolations, gives rise to more meaningful latent distributions, clusterings and more. We discuss related work in Sec. 6 and conclude the paper with an outlook in Sec. 7.

## 2 THE VARIATIONAL AUTOENCODERS ACTING AS THE GENERATOR

The variational autoencoder (VAE) proposed by Kingma & Welling (2014) is a simple yet powerful generative model which consists of two parts: (1) an *inference network* or *recognition network* (*encoder*) learns the latent representation (*codes*) of the data in the input space $\mathcal{X} = \mathbb{R}^D$; and (2) the *generator* (*decoder*) learns how to reconstruct the data from these latent space codes in $\mathcal{Z} = \mathbb{R}^d$.

Formally, a prior distribution is defined for the latent representations $p(\mathbf{z}) = \mathcal{N}(\mathbf{0}, \mathbb{I}_d)$, and there exists a mapping function $\boldsymbol{\mu}_\theta : \mathcal{Z} \rightarrow \mathcal{X}$ that generates a surface in $\mathcal{X}$. Moreover, we assume that another function $\boldsymbol{\sigma}_\theta : \mathcal{Z} \rightarrow \mathbb{R}_+^D$ captures the *error* (or *uncertainty*) between the actual data observation $\mathbf{x} \in \mathcal{X}$ and its reconstruction as $\mathbf{x} = \boldsymbol{\mu}_\theta(\mathbf{z}) + \boldsymbol{\sigma}_\theta \odot \boldsymbol{\epsilon}$, where $\boldsymbol{\epsilon} \sim \mathcal{N}(\mathbf{0}, \mathbb{I}_D)$ and $\odot$ is the Hadamard (element-wise) product. Then the likelihood is naturally defined as $p_\theta(\mathbf{x} \mid \mathbf{z}) = \mathcal{N}(\mathbf{x} \mid \boldsymbol{\mu}_\theta(\mathbf{z}), \mathbb{I}_D\boldsymbol{\sigma}_\theta^2(\mathbf{z}))$. The flexible functions $\boldsymbol{\mu}_\theta$ and $\boldsymbol{\sigma}_\theta$ are usually deep neural networks with parameters $\theta$.

However, the corresponding posterior distribution $p_\theta(\mathbf{z} \mid \mathbf{x})$ is unknown, as the marginal likelihood $p(\mathbf{x})$ is intractable. Hence, the posterior is approximated using a variational distribution $q_\phi(\mathbf{z} \mid \mathbf{x}) = \mathcal{N}(\mathbf{z} \mid \boldsymbol{\mu}_\phi(\mathbf{x}), \mathbb{I}_d\boldsymbol{\sigma}_\phi^2(\mathbf{x}))$, where the functions $\boldsymbol{\mu}_\phi : \mathcal{X} \rightarrow \mathcal{Z}$ and $\boldsymbol{\sigma}_\phi : \mathcal{X} \rightarrow \mathbb{R}_+^d$ are again deep neural networks with parameters $\phi$. Since the generator (decoder) is a composition of linear maps and activation functions, its smoothness is based solely on the chosen activation functions.

The optimal parameters $\theta$ and $\phi$ are found by maximizing the evidence lower bound (ELBO) of the marginal likelihood $p(\mathbf{x})$ as

$$\{\theta^*, \phi^*\} = \underset{\theta, \phi}{\operatorname{argmax}} \, \mathbb{E}_{q_\phi(\mathbf{z}|\mathbf{x})}[\log(p_\theta(\mathbf{x}|\mathbf{z}))] - \mathrm{KL}(q_\phi(\mathbf{z}|\mathbf{x})||p(\mathbf{z})), \tag{3}$$

where the bound follows from Jensen's inequality. The optimization is based on variations of gradient descent using the reparametrization trick (Kingma & Welling, 2014; Rezende et al., 2014). Further improvements have been proposed that provide more flexible posterior approximations (Rezende & Mohamed, 2015; Kingma et al., 2016) or tighter lower bound (Burda et al., 2016). In this paper, we consider the standard VAE for simplicity. The optimization problem in Eq. 3 is difficult since poor reconstructions by $\boldsymbol{\mu}_\theta$ can be explained by increasing the corresponding variance $\boldsymbol{\sigma}_\theta^2$. A common trick, which we also follow, is to optimize $\boldsymbol{\mu}_\theta$ while keeping $\boldsymbol{\sigma}_\theta^2$ constant, and then finally optimize for the variance $\boldsymbol{\sigma}_\theta^2$.

# 3 SURFACES AS THE FOUNDATION OF GENERATIVE MODELS

Mathematically, a *deterministic* generative model $\mathbf{x} = f(\mathbf{z})$ can be seen as a *surface model* (Gauss, 1827) if the generator $f$ is sufficiently smooth. Here, we briefly review the basic concepts on surfaces, as they form the mathematical foundation of this work.

Intuitively, a surface is a smoothly-connected set of points embedded in $\mathcal{X}$. When we want to make computations on a surface, it is often convenient to parametrize the surface by a low-dimensional (latent) variable $\mathbf{z}$ along with an appropriate function $f : \mathcal{Z} \to \mathcal{X}$. We let $d = \dim(\mathcal{Z})$ denote the intrinsic dimensionality of the surface, while $D = \dim(\mathcal{X})$ is the dimensionality of the input space. If we consider a smooth (latent) curve $\boldsymbol{\gamma}_t : [0,1] \to \mathcal{Z}$, then it has length $\int_0^1 \|\dot{\boldsymbol{\gamma}}_t\| \mathrm{d}t$, where $\dot{\boldsymbol{\gamma}}_t = \mathrm{d}\boldsymbol{\gamma}_t/\mathrm{d}t$ denotes the ve-

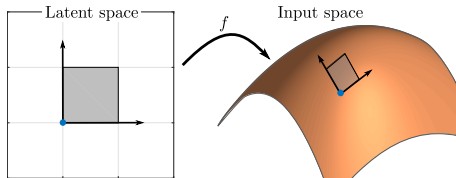

Figure 2: The Jacobian $\mathbf{J}$ of a nonlinear function $f$ provides a local basis in the input space, while $\sqrt{\det(\mathbf{J}^\intercal \mathbf{J})}$ measures the volume of an infinitesimal region.

locity of the curve. In practice, the low-dimensional parametrization $\mathcal{Z}$ often lacks a principled meaningful metric, so we measure lengths in input space by mapping the curve through $f$,

$$\mathrm{Length}[f(\boldsymbol{\gamma}_t)] = \int_0^1 \left\| \dot{f}(\boldsymbol{\gamma}_t) \right\|_2 \mathrm{d}t = \int_0^1 \left\| \mathbf{J}_{\boldsymbol{\gamma}_t} \dot{\boldsymbol{\gamma}}_t \right\|_2 \mathrm{d}t, \qquad \mathbf{J}_{\boldsymbol{\gamma}_t} = \left. \frac{\partial f}{\partial \mathbf{z}} \right|_{\mathbf{z} = \boldsymbol{\gamma}_t} \tag{4}$$

where the last step follows from the chain rule. This implies that the length of a curve $\boldsymbol{\gamma}_t$ along the surface can be computed directly in the latent space using the (locally defined) norm

$$\|\mathbf{J}_{\boldsymbol{\gamma}} \dot{\boldsymbol{\gamma}}\|_2 = \sqrt{(\mathbf{J}_{\boldsymbol{\gamma}} \dot{\boldsymbol{\gamma}})^\intercal (\mathbf{J}_{\boldsymbol{\gamma}} \dot{\boldsymbol{\gamma}})} = \sqrt{\dot{\boldsymbol{\gamma}}^\intercal (\mathbf{J}_{\boldsymbol{\gamma}}^\intercal \mathbf{J}_{\boldsymbol{\gamma}}) \dot{\boldsymbol{\gamma}}} = \sqrt{\dot{\boldsymbol{\gamma}}^\intercal \mathbf{M}_{\boldsymbol{\gamma}} \dot{\boldsymbol{\gamma}}}. \tag{5}$$

Here, $\mathbf{M}_{\boldsymbol{\gamma}} = \mathbf{J}_{\boldsymbol{\gamma}}^\intercal \mathbf{J}_{\boldsymbol{\gamma}}$ is a symmetric positive definite matrix, which acts akin to a local Mahalanobis distance measure. This gives rise to the definition of a *Riemannian metric*, which represents a smoothly changing inner product structure.

**Definition 1.** *A Riemannian metric* $\mathbf{M} : \mathcal{Z} \to \mathbb{R}^{d \times d}$ *is a smooth function that assigns a symmetric positive definite matrix to any point in* $\mathcal{Z}$.

It should be clear that if the generator function $f$ is sufficiently smooth, then $\mathbf{M}_{\boldsymbol{\gamma}}$ in Eq. 5 is a Riemannian metric.

When defining distances across a given surface, it is meaningful to seek the shortest curve connecting two points. Then a distance can be defined as the length of this curve. The shortest curve connecting points $\mathbf{z}_0$ and $\mathbf{z}_1$ is by (trivial) definition

$$\boldsymbol{\gamma}_t^{(\mathrm{shortest})} = \underset{\boldsymbol{\gamma}_t}{\operatorname{argmin}} \, \mathrm{Length}[f(\boldsymbol{\gamma}_t)], \qquad \boldsymbol{\gamma}_0 = \mathbf{z}_0, \ \boldsymbol{\gamma}_1 = \mathbf{z}_1. \tag{6}$$

A classic result of differential geometry (do Carmo, 1992) is that solutions to this optimization problem satisfy the following system of ordinary differential equations (ODEs)

$$\ddot{\boldsymbol{\gamma}}_t = -\frac{1}{2} \mathbf{M}_{\boldsymbol{\gamma}_t}^{-1} \left[ 2(\mathbb{I}_d \otimes \dot{\boldsymbol{\gamma}}_t^\intercal) \frac{\partial \mathrm{vec}\left[\mathbf{M}_{\boldsymbol{\gamma}_t}\right]}{\partial \boldsymbol{\gamma}_t} \dot{\boldsymbol{\gamma}}_t - \frac{\partial \mathrm{vec}\left[\mathbf{M}_{\boldsymbol{\gamma}_t}\right]}{\partial \boldsymbol{\gamma}_t}^\intercal (\dot{\boldsymbol{\gamma}}_t \otimes \dot{\boldsymbol{\gamma}}_t) \right], \tag{7}$$

where vec[·] stacks the columns of a matrix into a vector and $\otimes$ is the Kronecker product. For completeness, we provide a derivation of this result in Appendix A. Shortest curves can then be computed by solving the ODEs numerically; our implementation uses `bvp5c` from Matlab.

## 4 THE GEOMETRY OF STOCHASTIC GENERATORS

In the previous section, we considered *deterministic* generators $f$ to provide relevant background information. We now extend these results to the stochastic case; in particular we consider

$$f(\mathbf{z}) = \boldsymbol{\mu}(\mathbf{z}) + \boldsymbol{\sigma}(\mathbf{z}) \odot \boldsymbol{\epsilon}, \qquad \boldsymbol{\mu} : \mathcal{Z} \to \mathcal{X}, \ \boldsymbol{\sigma} : \mathcal{Z} \to \mathbb{R}_+^D, \ \boldsymbol{\epsilon} \sim \mathcal{N}(\mathbf{0}, \mathbb{I}_D). \tag{8}$$

This is the generator driving VAEs and related models. For our purposes, we will call $\boldsymbol{\mu}(\cdot)$ the *mean function* and $\boldsymbol{\sigma}^2(\cdot)$ the *variance function*.

Following the discussion from the previous section, it is natural to consider the Riemannian metric $\mathbf{M_z} = \mathbf{J_z^\intercal J_z}$ in the latent space. Since the generator is now stochastic, this metric also becomes stochastic, which complicates analysis. The following results, however, simplify matters.

**Theorem 1.** *If the stochastic generator in Eq. 8 has mean and variance functions that are at least twice differentiable, then the expected metric equals*

$$\overline{\mathbf{M}}_{\mathbf{z}} = \mathbb{E}_{p(\boldsymbol{\epsilon})}[\mathbf{M_z}] = \left(\mathbf{J}_{\mathbf{z}}^{(\boldsymbol{\mu})}\right)^\intercal \left(\mathbf{J}_{\mathbf{z}}^{(\boldsymbol{\mu})}\right) + \left(\mathbf{J}_{\mathbf{z}}^{(\boldsymbol{\sigma})}\right)^\intercal \left(\mathbf{J}_{\mathbf{z}}^{(\boldsymbol{\sigma})}\right), \tag{9}$$

*where $\mathbf{J}_{\mathbf{z}}^{(\boldsymbol{\mu})}$ and $\mathbf{J}_{\mathbf{z}}^{(\boldsymbol{\sigma})}$ are the Jacobian matrices of $\boldsymbol{\mu}(\cdot)$ and $\boldsymbol{\sigma}(\cdot)$.*

*Proof.* See Appendix B.

**Remark 1.** *By Definition 1, the metric tensor must change smoothly, which implies that the Jacobians must be smooth functions as well. This is easily ensured with activation functions for the neural networks that are $\mathcal{C}^2$ differentiable, e.g. $\tanh(\cdot)$, $\mathrm{sigmoid}(\cdot)$, and $\mathrm{softplus}(\cdot)$.*

**Theorem 2** (Due to Tosi et al. (2014)). *The variance of the metric under the $L_2$ measure vanishes when the data dimension goes to infinity, i.e. $\lim_{D\to\infty} \mathrm{Var}\left(\mathbf{M_z}\right) = 0$.*

Theorem 2 suggests that the (deterministic) expected metric $\overline{\mathbf{M}}_{\mathbf{z}}$ is a good approximation to the underlying stochastic metric when the data dimension is large. We make this approximation, which allows us to apply the theory of deterministic generators.

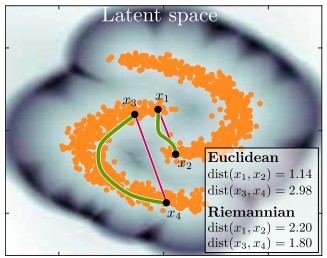

This expected metric has a particularly appealing form, where the two terms capture the distortion of the mean and the variance functions respectively. In particular, the variance term $(\mathbf{J}_{\mathbf{z}}^{(\boldsymbol{\sigma})})^\intercal(\mathbf{J}_{\mathbf{z}}^{(\boldsymbol{\sigma})})$ will be large in regions of the latent space, where the generator has large variance. This implies that induced distances will be large in regions of the latent space where the generator is highly uncertain, such that shortest paths will tend to avoid these regions. These paths

Figure 3: Example shortest paths and distances.

will then tend to follow the data in the latent space, c.f. Fig. 3. It is worth stressing, that no learning is needed to compute this metric: it only consists of terms that can be derived directly from the generator.

### 4.1 ENSURING PROPER GEOMETRY THROUGH MEANINGFUL VARIANCE FUNCTIONS

Theorem 1 informs us about how the geometry of the generative model depends on both the mean and the variance of the generator. Assuming successful training of the generator, we can expect to have good estimates of the geometry in regions near the data. But what happens in regions further away from the data? In general, the mean function cannot be expected to give useful extrapolations to such regions, so it is reasonable to require that the generator has high variance in regions that are not near the data. In practice, the neural net used to represent the variance function is only trained in regions where data is available, which implies that variance estimates are extrapolated to regions with no data. As neural nets tend to extrapolate poorly, *practical variance estimates tend to be arbitrarily poor in regions without data.*

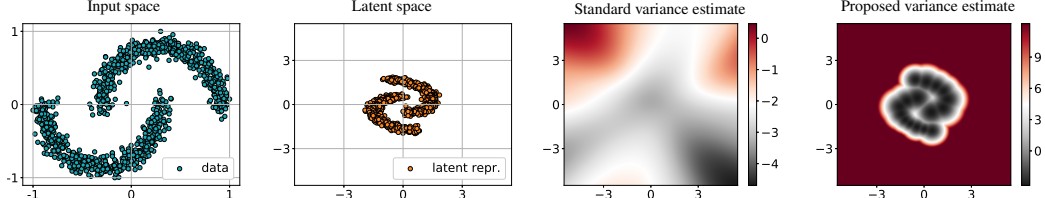

Figure 4: From left to right: training data in $\mathcal{X}$, latent representations in $\mathcal{Z}$, the standard deviation $\log(\sum_{j=1}^{D} \sigma_j(\mathbf{z}))$ for the standard variance network, and the proposed solution.

Figure 4 illustrates this problem. The first two panels show the data and its corresponding latent representations (here both input and latent dimensions are 2 to ease illustration). The third panel shows the variance function under a standard architecture, deep multilayer perceptron with *softplus* nonlinearity for the output layer. It is evident that variance estimates in regions without data are not representative of either uncertainty or error of the generative process; sometimes variance is high, sometimes it is low. From a probabilistic modeling point-of-view, this is disheartening. An informal survey of publicly available VAE implementations also reveals that it is common to enforce a constant unit variance everywhere; this is further disheartening.

For our purposes, we need well-behaved variance functions to ensure a well-behaved geometry, but reasonable variance estimates are of general use. Here, as a general strategy, we propose to model the inverse variance with a network that extrapolates towards zero. This at least ensures that variances are large in regions without data. Specifically, we model the *precision* as $\boldsymbol{\beta}_\psi(\mathbf{z}) = \frac{1}{\boldsymbol{\sigma}_\psi^2(\mathbf{z})}$, where all operations are element-wise. Then, we model this precision with a *radial basis function (RBF) neural network* (Que & Belkin, 2016). Formally this is written

$$\boldsymbol{\beta}_\psi(\mathbf{z}) = \mathbf{W}\mathbf{v}(\mathbf{z}) + \boldsymbol{\zeta}, \quad \text{with} \quad v_k(\mathbf{z}) = \exp\left(-\lambda_k \|\mathbf{z} - \mathbf{c}_k\|_2^2\right), \ k = 1, \dots, K, \qquad (10)$$

where $\psi$ are all parameters, $\mathbf{W} \in \mathbb{R}_{>0}^{D \times K}$ are the *positive* weights of the network (positivity ensures a positive precision), $\mathbf{c}_k$ and $\lambda_k$ are the centers and the bandwidth of the $K$ radial basis functions, and $\boldsymbol{\zeta} \to \mathbf{0}$ is a vector of positive constants to prevent division by zero. It is easy to see that with this approach the variance of the generator increases with the distance to the centers. The right-most panel of Fig. 4 shows an estimated variance function, which indeed has the desired property that variance is large outside the data support. Further, note the increased variance between the two clusters, which captures that even interpolating between clusters comes with a level of uncertainty. In Appendix C we also demonstrate that this simple variance model improves the marginal likelihood $p(\mathbf{x})$ on held-out data.

Training the variance network amounts to fitting the RBF network. Assuming we have already trained the inference network (Sec. 2), we can encode the training data, and use $k$-means to estimate the RBF centers. Then, an estimate for the bandwidths of each kernel can be computed as

$$\lambda_k = \frac{1}{2}\left(a\frac{1}{|\mathcal{C}_k|}\sum_{\mathbf{z}_j \in \mathcal{C}_k} \|\mathbf{z}_j - \mathbf{c}_k\|_2\right)^{-2} \qquad (11)$$

where the hyper-parameter $a \in \mathbb{R}_+$ controls the curvature of the Riemannian metric, i.e. how fast it changes based on the uncertainty. Since the mean function of the generator is already trained, the weights of the RBF can be found using projected gradient descent to ensure positive weights.

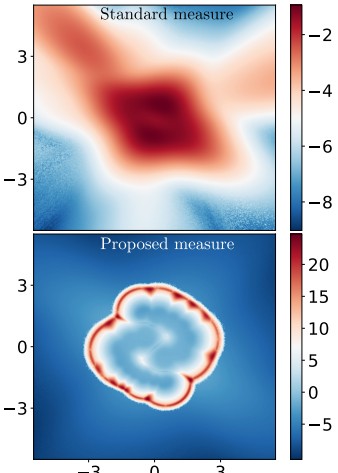

Figure 5: Comparison of (log) measures of standard (top) and proposed (bottom) variances.

One visualization of the distortion of the latent space relative to the input space is the geometric *volume measure* $\sqrt{\det(\mathbf{M_z})}$, which captures the volume of an infinitesimal area in the input space. Figure 5 shows this volume measure for both standard variance functions as well as our proposed RBF model. We see that the proposed model captures the trend of the data, unlike the standard model.

## 5 EMPIRICAL RESULTS

We demonstrate the usefulness of the geometric view of the latent space with several experiments. Model and implementation details can be found in Appendix D. In all experiments we first train a VAE and then use the induced Riemannian metric.

### 5.1 MEANINGFUL DISTANCES

First we seek to quantify if the induced Riemannian distance in the latent space is more useful than the usual Euclidean distance. For this we perform basic $k$-means clustering under the two metrics. We construct 3 sets of MNIST digits, using 1000 random samples for each digit. We train a VAE for

| Digits | Linear | Riemannian |
|--------|--------|------------|
| $\{0, 1, 2\}$ | $77.57(\pm0.87)\%$ | $\mathbf{94.28(\pm1.14)\%}$ |
| $\{3, 4, 7\}$ | $77.80(\pm0.91)\%$ | $\mathbf{89.54(\pm1.61)\%}$ |
| $\{5, 6, 9\}$ | $64.93(\pm0.96)\%$ | $\mathbf{81.13(\pm2.52)\%}$ |

Table 1: The $F$-measure results for $k$-means.

each set, and then subdivide each into 10 sub-sets, and performed $k$-means clustering under both distances. One example result is shown in Fig. 6. Here it is evident that, since the latent points roughly follow a unit Gaussian, there is little structure to be discovered by the Euclidean $k$-means, and consequently it performs poorly. The Riemannian clustering is remarkably accurate. Summary statistics across all subsets are provided in Table 1, which shows the established $F$-measure for clustering accuracy. Again, the Riemannian metric significantly improves clustering. This implies that the underlying Riemannian distance is more useful than its Euclidean counterpart.

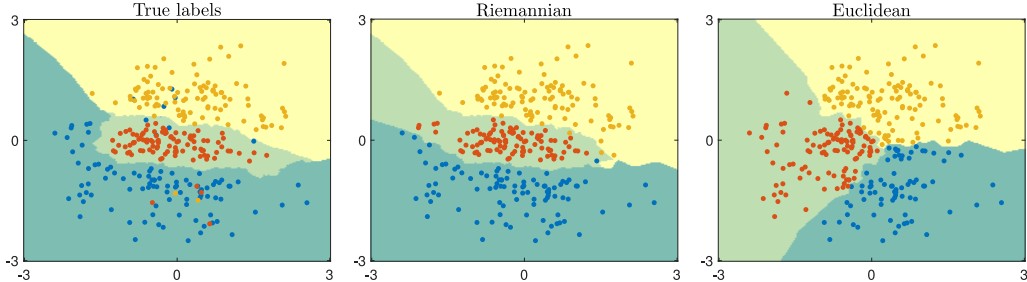

Figure 6: The result of $k$-means comparing the distance measures. For the decision boundaries we used 7-NN classification.

### 5.2 INTERPOLATIONS

Next, we investigate whether the Riemannian metric gives more meaningful interpolations. First, we train a VAE for the digits 0 and 1 from MNIST. The upper left panel of Fig. 7 shows the latent space with the Riemannian measure as background color, together with two interpolations. Images generated by both Riemannian and Euclidean interpolations are shown in the bottom of Fig. 7. The Euclidean interpolations seem to have a very abrupt change when transitioning from one class to another. The Riemannian interpolant gives smoother changes in the generated images. The top-right panel of the figure shows the auto-correlation of images along the interpolants; again we see a very abrupt change in the Euclidean interpolant, while the Riemannian is significantly smoother. We also train a convolutional VAE on frames from a video. Figure 8 shows the corresponding latent space and some sample interpolations. As before, we see more smooth changes in generated images when we take the Riemannian metric into account.

### 5.3 LATENT PROBABILITY DISTRIBUTIONS

We have seen strong indications that the Riemannian metric gives a more meaningful view of the latent space, which may also improve probability distributions in the latent space. A relevant candidate distribution is the *locally adaptive normal distribution (LAND)* (Arvanitidis et al., 2016)

$$\text{LAND}(\mathbf{z} \mid \boldsymbol{\mu}, \boldsymbol{\Sigma}) \propto \exp\left(-\frac{1}{2}\text{dist}_{\boldsymbol{\Sigma}}^2(\mathbf{z}, \boldsymbol{\mu})\right), \tag{12}$$

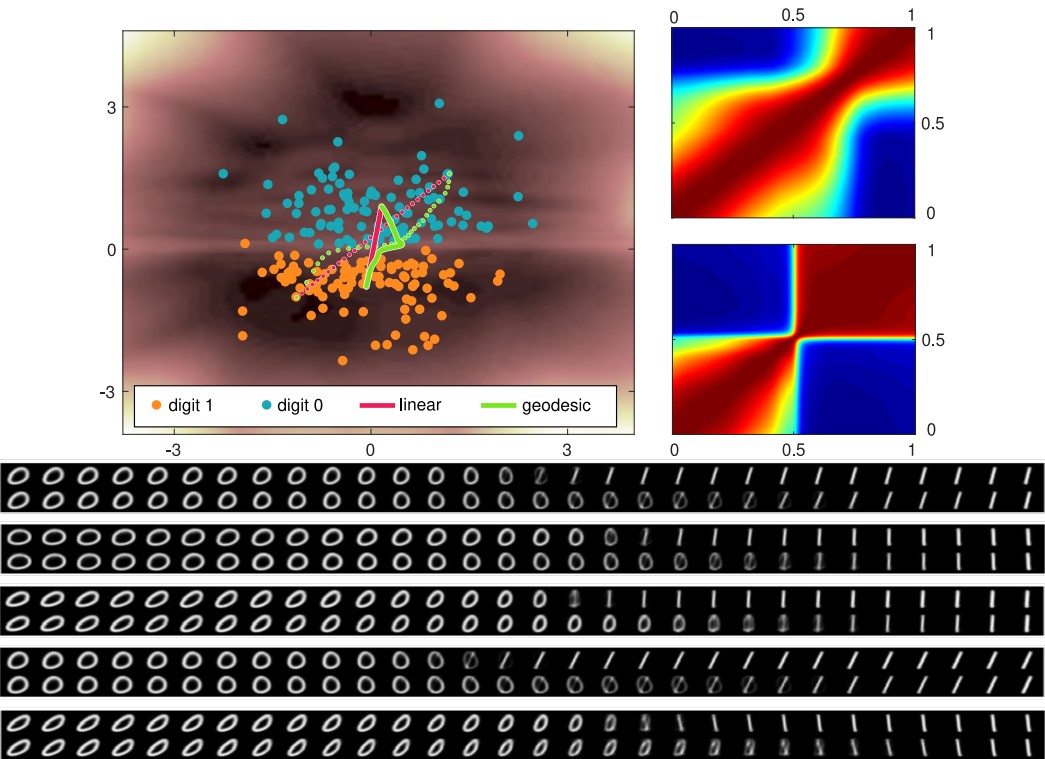

Figure 7: *Left:* the latent space with example interpolants. *Right:* auto-correlations of Riemannian (top) and Euclidean (bottom) samples along the curves of the left panel. *Bottom:* decoded images along Euclidean (top rows) and Riemannian (bottom rows) interpolants.

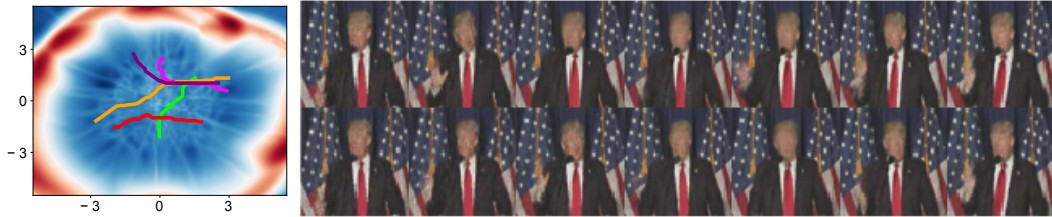

Figure 8: *Left:* the latent space and geodesic interpolants. *Right:* samples comparing Euclidean (top row) with Riemannian (bottom row) interpolation. Corresponding videos can be found here.

where $\mathrm{dist}_{\mathbf{\Sigma}}$ is the Riemannian extension of Mahalanobis distance. We fit a mixture of two LANDs to the MNIST data from Sec. 5.2 alongside a mixture of Euclidean normal distributions. The first column of Fig. 9 shows the density functions of the two mixture models. Only the Riemannian model reveals the underlying clusters. We then sample 40 points from each component of these generative models[1] (center column of the figure). We see that the Riemannian model generates high-quality samples, whereas the Euclidean model generates several samples in regions where the generator is not trained and therefore produces blurry images. Finally, the right column of Fig. 9 shows all pairwise distances between the latent points under both Riemannian and Euclidean distances. Again, we see that the geometric view clearly reveals the underlying clusters.

## 5.4  RANDOM WALK ON THE DATA MANIFOLD

Finally, we consider random walks over the data manifold, which is a common tool for exploring latent spaces. To avoid the walk drifting outside the data support, practical implementations artificially

---

[1]We do not follow common practice and sort samples by their likelihood, as this hides low-quality samples.

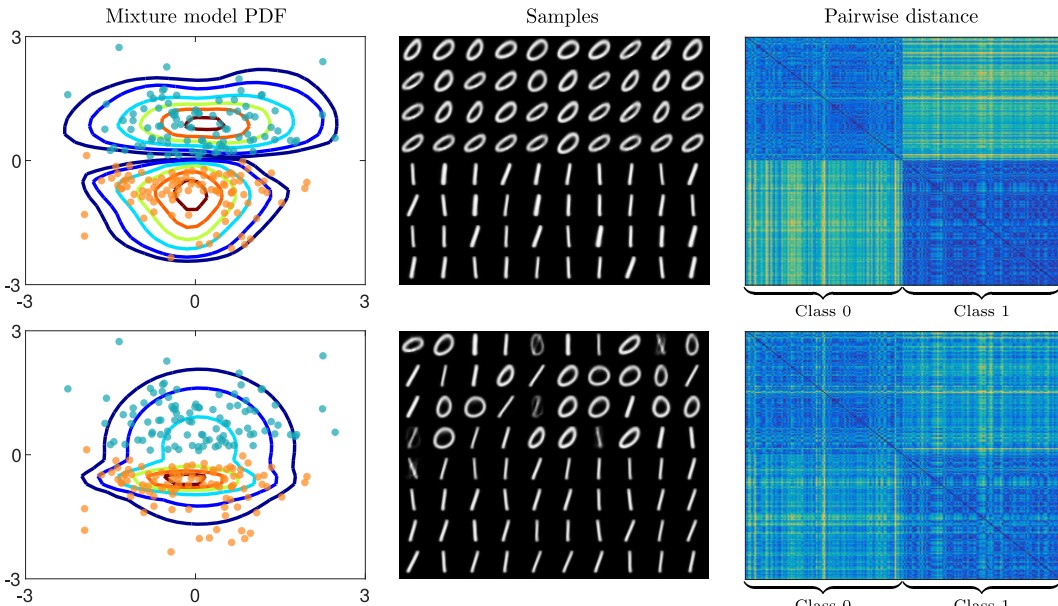

Figure 9: From *left* to *right*: the mixture models, generated samples, and pairwise distances. *Top* row corresponds to the Riemannian model and *bottom* row to the Euclidean model.

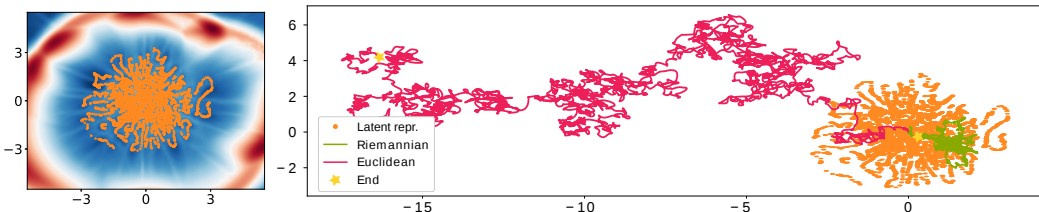

Figure 10: *Left:* the measure in the latent space. *Right:* the random walks.

restrict the walk to stay inside the $[-1, 1]^d$ hypercube. Here, we consider unrestricted Brownian motion under both the Euclidean and Riemannian metric. We perform this random walk in the latent space of the convolutional VAE from Sec. 5.2. Figure 10 shows example walks, while Fig. 11 shows generated images (video here). While the Euclidean random walk moves freely, the Riemannian walk stays within the support of the data. This is explained in the left panel of Fig. 10, which shows that the variance term in the Riemannian metric creates a "wall" around the data, which the random walk will only rarely cross. These "walls" also force shortest paths to follow the data.

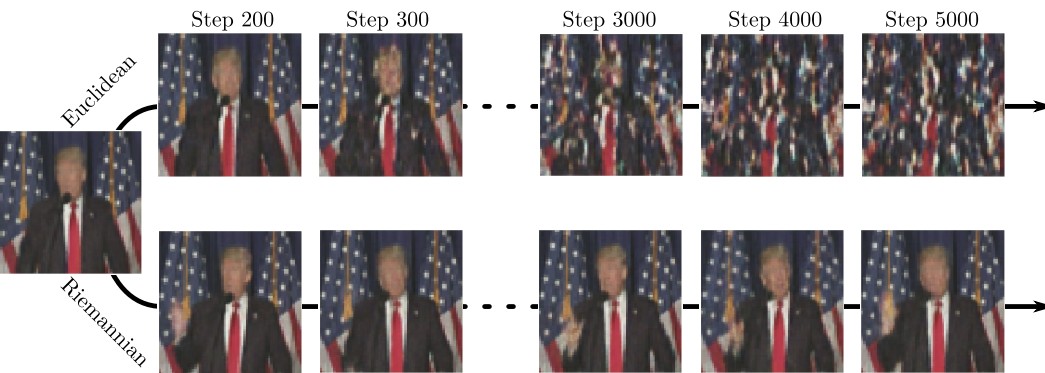

Figure 11: The comparison of the random walks, at the steps 200, 300, 3000, 4000 and 5000.

## 6    RELATED WORK

**Generative models.**    This unsupervised learning category attracted a lot of attention, especially, due to the advances on the deep neural networks. We have considered VAEs (Kingma & Welling, 2014; Rezende et al., 2014), but the ideas extend to similar related models. These include extensions that provide more flexible approximate posteriors (Rezende & Mohamed, 2015; Kingma et al., 2016). GANs (Goodfellow et al., 2014) also fall in this category, as these models have an explicit generator. While the inference network is not a necessary component in the GAN model, it has been shown that incorporating it improves overall performance (Donahue et al., 2017; Dumoulin et al., 2017). The same thoughts hold for approaches that transform the latent space through a sequence of bijective functions (Dinh et al., 2017)

**Geometry in neural networks.**    Bengio et al. (2013) discuss the importance of geometry in neural networks as a tool to understand local generalization. For instance, the Jacobian matrix is a measure of smoothness for a function that interpolates a surface to the given data. This is exactly the implication in (Rifai et al., 2011), where the norm of the Jacobian acts as a regularizer for the deterministic autoencoder. Recently, Kumar et al. (2017) used the Jacobian to inject invariances in a classifier.

**Riemannian Geometry.**    Like the present paper, Tosi et al. (2014) derive a suitable Riemannian metric in Gaussian process (GP) latent variable models (Lawrence, 2005), but the computational complexity of GPs causes practical concerns. Unlike works that explicitly learn a Riemannian metric (Hauberg et al., 2012; Peltonen et al., 2004), our metric is fully derived from the generator and requires no extra learning once the generator is available.

## 7    DISCUSSION AND FURTHER EXTENSIONS

The geometric interpretation of representation learning is that the latent space is a compressed and flattened version of the data manifold. We show that the *actual* geometry of the data manifold can be more complex than it first appears.

Here we have initiated the study of proper geometries for generative models. We showed that the latent space not only provides a low-dimensional representation of the data manifold, but at the same time, can reveal the underlying geometrical structure. We proposed a new variance network for the generator, which provides meaningful uncertainty estimates while regularizing the geometry. The new detailed understanding of the geometry provides us with more relevant distance measures, as demonstrated by the fact that a $k$-means clustering, on these distances, is better aligned with the ground truth label structure than a clustering based on conventional Euclidean distances. We also found that the new distance measure produces smoother interpolation, and when training Riemannian "LAND" mixture models based on the new geometry, the components aligned much better with the ground truth group structure. Finally, inspired by the recent interest in sequence generation by random walks in latent space, we found that geometrically informed random walks stayed on the manifold for much longer runs than sequences based on Euclidean random walks.

The presented analysis easily extends to sophisticated generative models, where the latent space will be potentially endowed with more flexible nonlinear structures. This directly implies particularly interesting geometrical models. An obvious question is: can the geometry of the latent space play a role while we learn the generative model? Either way, we believe that this geometric perspective provides a new way of thinking and further interpreting the generative models, while at the same time it encourages development of new nonlinear models in the representation space.

### ACKNOWLEDGMENTS

LKH is supported by Innovation Fund Denmark / the Danish Center for Big Data Analytics Driven Innovation. SH was supported by a research grant (15334) from VILLUM FONDEN. This project has received funding from the European Research Council (ERC) under the European Union's Horizon 2020 research and innovation programme (grant agreement n$^o$ 757360). We gratefully acknowledge the support of the NVIDIA Corporation with the donation of the used Titan Xp GPU.

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

## A    THE DERIVATION OF THE GEODESIC DIFFERENTIAL EQUATION

The shortest path between two points $\mathbf{x}, \mathbf{y} \in \mathcal{M}$ on a Riemannian manifold $\mathcal{M}$ is found by optimizing the functional

$$\boldsymbol{\gamma}_t^{(\text{shortest})} = \underset{\boldsymbol{\gamma}_t}{\operatorname{argmin}} \int_0^1 \sqrt{\langle \dot{\boldsymbol{\gamma}}_t, \mathbf{M}_{\boldsymbol{\gamma}_t} \dot{\boldsymbol{\gamma}}_t \rangle} \mathrm{d}t, \quad \boldsymbol{\gamma}(0) = \mathbf{x}, \ \boldsymbol{\gamma}(1) = \mathbf{y} \tag{13}$$

where $\boldsymbol{\gamma}_t : [0,1] \to \mathcal{M}$ and $\dot{\boldsymbol{\gamma}}_t = \frac{\partial \boldsymbol{\gamma}_t}{\partial t}$. The minima of this problem can be found instead by optimizing the *curve energy* (do Carmo, 1992), so the functional becomes

$$\boldsymbol{\gamma}_t^{(\text{shortest})} = \underset{\boldsymbol{\gamma}_t}{\operatorname{argmin}} \int_0^1 \langle \dot{\boldsymbol{\gamma}}_t, \mathbf{M}_{\boldsymbol{\gamma}_t} \dot{\boldsymbol{\gamma}}_t \rangle \mathrm{d}t, \quad \boldsymbol{\gamma}(0) = \mathbf{x}, \ \boldsymbol{\gamma}(1) = \mathbf{y}. \tag{14}$$

The inner product can be written explicitly as

$$L(\boldsymbol{\gamma}_t, \dot{\boldsymbol{\gamma}}_t, \mathbf{M}_{\boldsymbol{\gamma}_t}) = \langle \dot{\boldsymbol{\gamma}}_t, \mathbf{M}_{\boldsymbol{\gamma}_t} \dot{\boldsymbol{\gamma}}_t \rangle = \sum_{i=1}^d \sum_{j=1}^d \dot{\gamma}_t^{(i)} \cdot \dot{\gamma}_t^{(j)} \cdot M_{\boldsymbol{\gamma}_t}^{(ij)} = (\dot{\boldsymbol{\gamma}}_t \otimes \dot{\boldsymbol{\gamma}}_t)^\mathsf{T} \mathrm{vec}\big[\mathbf{M}_{\boldsymbol{\gamma}_t}\big] \tag{15}$$

where the index in the parenthesis represents the corresponding element in the vector or matrix. In the derivation the $\otimes$ is the usual Kronecker product and the $\mathrm{vec}[\cdot]$ stacks the column of a matrix into a vector. We find the minimizers by the Euler-Lagrange equation

$$\frac{\partial L}{\partial \boldsymbol{\gamma}_t} = \frac{\partial}{\partial t} \frac{\partial L}{\partial \dot{\boldsymbol{\gamma}}_t} \tag{16}$$

where

$$\frac{\partial}{\partial t} \frac{\partial L}{\partial \dot{\boldsymbol{\gamma}}_t} = \frac{\partial}{\partial t} \frac{\partial \langle \dot{\boldsymbol{\gamma}}_t, \mathbf{M}_{\boldsymbol{\gamma}_t} \dot{\boldsymbol{\gamma}}_t \rangle}{\partial \dot{\boldsymbol{\gamma}}_t} = \frac{\partial}{\partial t} \left( 2 \cdot \mathbf{M}_{\boldsymbol{\gamma}_t} \dot{\boldsymbol{\gamma}}_t \right) = 2 \left[ \frac{\partial \mathbf{M}_{\boldsymbol{\gamma}_t}}{\partial t} \dot{\boldsymbol{\gamma}}_t + \mathbf{M}_{\boldsymbol{\gamma}_t} \ddot{\boldsymbol{\gamma}}_t \right]. \tag{17}$$

Since the term

$$\frac{\partial \mathbf{M}_{\boldsymbol{\gamma}_t}}{\partial t} = \begin{bmatrix} \frac{\partial M_{\boldsymbol{\gamma}_t}^{(11)}}{\partial t} & \cdots & \frac{\partial M_{\boldsymbol{\gamma}_t}^{(1D)}}{\partial t} \\ \frac{\partial M_{\boldsymbol{\gamma}_t}^{(21)}}{\partial t} & \cdots & \frac{\partial M_{\boldsymbol{\gamma}_t}^{(2D)}}{\partial t} \\ \vdots & \ddots & \vdots \\ \frac{\partial M_{\boldsymbol{\gamma}}^{(D1)}}{\partial t} & \cdots & \frac{\partial M_{\boldsymbol{\gamma}_t}^{(DD)}}{\partial t} \end{bmatrix} = \begin{bmatrix} \frac{\partial M_{\boldsymbol{\gamma}_t}^{(11)}}{\partial \boldsymbol{\gamma}_t}^\mathsf{T} \dot{\boldsymbol{\gamma}}_t & \cdots & \frac{\partial M_{\boldsymbol{\gamma}_t}^{(1D)}}{\partial \boldsymbol{\gamma}_t}^\mathsf{T} \dot{\boldsymbol{\gamma}}_t \\ \frac{\partial M_{\boldsymbol{\gamma}_t}^{(21)}}{\partial \boldsymbol{\gamma}_t}^\mathsf{T} \dot{\boldsymbol{\gamma}}_t & \cdots & \frac{\partial M_{\boldsymbol{\gamma}}^{(2D)}}{\partial \boldsymbol{\gamma}_t}^\mathsf{T} \dot{\boldsymbol{\gamma}}_t \\ \vdots & \ddots & \vdots \\ \frac{\partial M_{\boldsymbol{\gamma}_t}^{(D1)}}{\partial \boldsymbol{\gamma}_t}^\mathsf{T} \dot{\boldsymbol{\gamma}}_t & \cdots & \frac{\partial M_{\boldsymbol{\gamma}_t}^{(DD)}}{\partial \boldsymbol{\gamma}_t}^\mathsf{T} \dot{\boldsymbol{\gamma}}_t \end{bmatrix} \tag{18}$$

we can write the right hand side of the Eq. 16 as

$$\frac{\partial}{\partial t} \frac{\partial L}{\partial \dot{\boldsymbol{\gamma}}_t} = 2 \left[ (\mathbb{I}_d \otimes \dot{\boldsymbol{\gamma}}_t^\mathsf{T}) \frac{\partial \mathrm{vec}\big[\mathbf{M}_{\boldsymbol{\gamma}_t}\big]}{\partial \boldsymbol{\gamma}_t} \dot{\boldsymbol{\gamma}}_t + \mathbf{M}_{\boldsymbol{\gamma}_t} \ddot{\boldsymbol{\gamma}}_t \right]. \tag{19}$$

The left hand side term of the Eq. 16 is equal to

$$\frac{\partial L}{\partial \boldsymbol{\gamma}_t} = \frac{\partial}{\partial \boldsymbol{\gamma}_t} \left( (\dot{\boldsymbol{\gamma}}_t \otimes \dot{\boldsymbol{\gamma}}_t)^\mathsf{T} \mathrm{vec}\big[\mathbf{M}_{\boldsymbol{\gamma}_t}\big] \right) = (\dot{\boldsymbol{\gamma}}_t \otimes \dot{\boldsymbol{\gamma}}_t)^\mathsf{T} \frac{\partial \mathrm{vec}\big[\mathbf{M}_{\boldsymbol{\gamma}_t}\big]}{\partial \boldsymbol{\gamma}_t}. \tag{20}$$

The final system of 2$^{\text{nd}}$ order ordinary differential equations is

$$\ddot{\boldsymbol{\gamma}}_t = -\frac{1}{2} \mathbf{M}_{\boldsymbol{\gamma}_t}^{-1} \left[ 2 \cdot (\mathbb{I}_d \otimes \dot{\boldsymbol{\gamma}}_t^\mathsf{T}) \frac{\partial \mathrm{vec}\big[\mathbf{M}_{\boldsymbol{\gamma}_t}\big]}{\partial \boldsymbol{\gamma}_t} \dot{\boldsymbol{\gamma}}_t - \frac{\partial \mathrm{vec}\big[\mathbf{M}_{\boldsymbol{\gamma}_t}\big]}{\partial \boldsymbol{\gamma}_t}^\mathsf{T} (\dot{\boldsymbol{\gamma}}_t \otimes \dot{\boldsymbol{\gamma}}_t) \right]. \tag{21}$$

## B    THE DERIVATION OF THE RIEMANNIAN METRIC

The proof of Theorem 1.

*Proof.* As we introduced in Eq. 8 the stochastic generator is

$$f(\mathbf{z}) = \boldsymbol{\mu}(\mathbf{z}) + \boldsymbol{\sigma}(\mathbf{z}) \odot \boldsymbol{\epsilon}, \qquad \boldsymbol{\mu} : \mathcal{Z} \to \mathcal{X}, \ \boldsymbol{\sigma} : \mathcal{Z} \to \mathbb{R}^D_+, \ \boldsymbol{\epsilon} \sim \mathcal{N}(\mathbf{0}, \mathbb{I}_D). \tag{22}$$

Thus, we can compute the corresponding Jacobian as follows

$$\frac{\partial f(\mathbf{z})}{\partial \mathbf{z}} = \mathbf{J}_{\mathbf{z}} = \begin{bmatrix} \frac{\partial f_{\mathbf{z}}^{(1)}}{\partial z_1} & \frac{\partial f_{\mathbf{z}}^{(1)}}{\partial z_2} & \cdots & \frac{\partial f_{\mathbf{z}}^{(1)}}{\partial z_d} \\ \frac{\partial f_{\mathbf{z}}^{(2)}}{\partial z_1} & \frac{\partial f_{\mathbf{z}}^{(2)}}{\partial z_2} & \cdots & \frac{\partial f_{\mathbf{z}}^{(2)}}{\partial z_d} \\ \vdots & \vdots & \ddots & \vdots \\ \frac{\partial f_{\mathbf{z}}^{(D)}}{\partial z_1} & \frac{\partial f_{\mathbf{z}}^{(D)}}{\partial z_2} & \cdots & \frac{\partial f_{\mathbf{z}}^{(D)}}{\partial z_d} \end{bmatrix}_{D \times d} \tag{23}$$

$$= \underbrace{\begin{bmatrix} \frac{\partial \boldsymbol{\mu}_{\mathbf{z}}^{(1)}}{\partial z_1} & \frac{\partial \boldsymbol{\mu}_{\mathbf{z}}^{(1)}}{\partial z_2} & \cdots & \frac{\partial \boldsymbol{\mu}_{\mathbf{z}}^{(1)}}{\partial z_d} \\ \frac{\partial \boldsymbol{\mu}_{\mathbf{z}}^{(2)}}{\partial z_1} & \frac{\partial \boldsymbol{\mu}_{\mathbf{z}}^{(2)}}{\partial z_2} & \cdots & \frac{\partial \boldsymbol{\mu}_{\mathbf{z}}^{(2)}}{\partial z_d} \\ \vdots & \vdots & \ddots & \vdots \\ \frac{\partial \boldsymbol{\mu}_{\mathbf{z}}^{(D)}}{\partial z_1} & \frac{\partial \boldsymbol{\mu}_{\mathbf{z}}^{(D)}}{\partial z_2} & \cdots & \frac{\partial \boldsymbol{\mu}_{\mathbf{z}}^{(D)}}{\partial z_d} \end{bmatrix}_{D \times d}}_{\mathbf{A}} + \underbrace{[\mathbf{S}_1 \boldsymbol{\epsilon}, \ \mathbf{S}_2 \boldsymbol{\epsilon}, \ \cdots, \ \mathbf{S}_d \boldsymbol{\epsilon}]_{D \times d}}_{\mathbf{B}}, \tag{24}$$

$$\text{where} \quad \mathbf{S}_i = \begin{bmatrix} \frac{\partial \sigma_{\mathbf{z}}^{(1)}}{\partial z_i} & 0 & \cdots & 0 \\ 0 & \frac{\partial \sigma_{\mathbf{z}}^{(2)}}{\partial z_i} & \cdots & 0 \\ \vdots & \vdots & \ddots & \vdots \\ 0 & 0 & \cdots & \frac{\partial \sigma_{\mathbf{z}}^{(D)}}{\partial z_i} \end{bmatrix}_{D \times D}, \quad i = 1, \ldots, d \tag{25}$$

and the resulting "random" metric in the latent space is $\mathbf{M}_{\mathbf{z}} = \mathbf{J}_{\mathbf{z}}^{\intercal} \mathbf{J}_{\mathbf{z}}$. The randomness is due to the random variable $\boldsymbol{\epsilon}$, and thus, we can compute the expectation

$$\overline{\mathbf{M}}_{\mathbf{z}} = \mathbb{E}_{p(\boldsymbol{\epsilon})}[\mathbf{M}_{\mathbf{z}}] = \mathbb{E}_{p(\boldsymbol{\epsilon})}[(\mathbf{A} + \mathbf{B})^{\intercal}(\mathbf{A} + \mathbf{B})] = \mathbb{E}_{p(\boldsymbol{\epsilon})}[\mathbf{A}^{\intercal}\mathbf{A} + \mathbf{A}^{\intercal}\mathbf{B} + \mathbf{B}^{\intercal}\mathbf{A} + \mathbf{B}^{\intercal}\mathbf{B}]. \tag{26}$$

Using the linearity of expectation we get that

$$\mathbb{E}_{p(\boldsymbol{\epsilon})}[\mathbf{A}^{\intercal}\mathbf{B}] = \mathbb{E}_{p(\boldsymbol{\epsilon})}\left[\mathbf{A}^{\intercal}[\mathbf{S}_1\boldsymbol{\epsilon}, \mathbf{S}_2\boldsymbol{\epsilon}, \cdots, \mathbf{S}_d\boldsymbol{\epsilon}]\right] = \mathbf{A}^{\intercal}[\mathbf{S}_1\underbrace{\mathbb{E}_{p(\boldsymbol{\epsilon})}[\boldsymbol{\epsilon}]}_{\mathbf{0}}, \ldots, \mathbf{0}] = 0 \tag{27}$$

because $\mathbb{E}_{p(\boldsymbol{\epsilon})}[\boldsymbol{\epsilon}] = \mathbf{0}$. The other term

$$\mathbb{E}_{p(\boldsymbol{\epsilon})}[\mathbf{B}^{\intercal}\mathbf{B}] = \mathbb{E}_{p(\boldsymbol{\epsilon})}\left( \begin{bmatrix} \boldsymbol{\epsilon}^{\intercal}\mathbf{S}_1 \\ \boldsymbol{\epsilon}^{\intercal}\mathbf{S}_2 \\ \vdots \\ \boldsymbol{\epsilon}^{\intercal}\mathbf{S}_d \end{bmatrix}_{d \times D} [\mathbf{S}_1\boldsymbol{\epsilon}, \mathbf{S}_2\boldsymbol{\epsilon}, \cdots, \mathbf{S}_d\boldsymbol{\epsilon}] \right) \tag{28}$$

$$= \mathbb{E}_{p(\boldsymbol{\epsilon})}\left( \begin{bmatrix} \boldsymbol{\epsilon}^{\intercal}\mathbf{S}_1\mathbf{S}_1\boldsymbol{\epsilon} & \boldsymbol{\epsilon}^{\intercal}\mathbf{S}_1\mathbf{S}_2\boldsymbol{\epsilon} & \cdots & \boldsymbol{\epsilon}^{\intercal}\mathbf{S}_1\mathbf{S}_d\boldsymbol{\epsilon} \\ \boldsymbol{\epsilon}^{\intercal}\mathbf{S}_2\mathbf{S}_1\boldsymbol{\epsilon} & \boldsymbol{\epsilon}^{\intercal}\mathbf{S}_2\mathbf{S}_2\boldsymbol{\epsilon} & \cdots & \boldsymbol{\epsilon}^{\intercal}\mathbf{S}_2\mathbf{S}_d\boldsymbol{\epsilon} \\ \vdots & \vdots & \vdots & \vdots \\ \boldsymbol{\epsilon}^{\intercal}\mathbf{S}_d\mathbf{S}_1\boldsymbol{\epsilon} & \boldsymbol{\epsilon}^{\intercal}\mathbf{S}_d\mathbf{S}_2\boldsymbol{\epsilon} & \cdots & \boldsymbol{\epsilon}^{\intercal}\mathbf{S}_d\mathbf{S}_d\boldsymbol{\epsilon} \end{bmatrix} \right) \tag{29}$$

$$\text{with} \quad \mathbb{E}_{p(\boldsymbol{\epsilon})}[\boldsymbol{\epsilon}^{\intercal}\mathbf{S}_i\mathbf{S}_j\boldsymbol{\epsilon}] = \mathbb{E}_{p(\boldsymbol{\epsilon})}\left[ \left( \epsilon_1\frac{\partial\sigma_{\mathbf{z}}^{(1)}}{\partial z_i}, \epsilon_2\frac{\partial\sigma_{\mathbf{z}}^{(2)}}{\partial z_i}, \cdots, \epsilon_D\frac{\partial\sigma_{\mathbf{z}}^{(D)}}{\partial z_i} \right) \begin{pmatrix} \epsilon_1\frac{\partial\sigma_{\mathbf{z}}^{(1)}}{\partial z_j} \\ \epsilon_2\frac{\partial\sigma_{\mathbf{z}}^{(2)}}{\partial z_j} \\ \vdots \\ \epsilon_D\frac{\partial\sigma_{\mathbf{z}}^{(D)}}{\partial z_j} \end{pmatrix} \right] \tag{30}$$

$$= \mathbb{E}_{p(\boldsymbol{\epsilon})}\left[ \epsilon_1^2 \left( \frac{\partial\sigma_{\mathbf{z}}^{(1)}}{\partial z_i}\frac{\partial\sigma_{\mathbf{z}}^{(1)}}{\partial z_j} \right) + \epsilon_2^2 \left( \frac{\partial\sigma_{\mathbf{z}}^{(2)}}{\partial z_i}\frac{\partial\sigma_{\mathbf{z}}^{(2)}}{\partial z_j} \right) + \cdots \epsilon_D^2 \left( \frac{\partial\sigma_{\mathbf{z}}^{(D)}}{\partial z_i}\frac{\partial\sigma_{\mathbf{z}}^{(D)}}{\partial z_j} \right) \right] \tag{31}$$

$$= diag(\mathbf{S}_i)^{\intercal} diag(\mathbf{S}_j), \tag{32}$$

because $\mathbb{E}_{p(\epsilon)}[\epsilon_i^2] = 1$, $\forall i = 1, \dots, D$.

The matrix $\mathbf{A} = \mathbf{J}_{\mathbf{z}}^{(\boldsymbol{\mu})}$ and for the variance network

$$\mathbf{J}_{\mathbf{z}}^{(\boldsymbol{\sigma})} = \begin{bmatrix} \frac{\partial \boldsymbol{\sigma}_{\mathbf{z}}^{(1)}}{\partial z_1} & \frac{\partial \boldsymbol{\sigma}_{\mathbf{z}}^{(1)}}{\partial z_2} & \cdots & \frac{\partial \boldsymbol{\sigma}_{\mathbf{z}}^{(1)}}{\partial z_d} \\ \frac{\partial \boldsymbol{\sigma}_{\mathbf{z}}^{(2)}}{\partial z_1} & \frac{\partial \boldsymbol{\sigma}_{\mathbf{z}}^{(2)}}{\partial z_2} & \cdots & \frac{\partial \boldsymbol{\sigma}_{\mathbf{z}}^{(2)}}{\partial z_d} \\ \vdots & \vdots & \ddots & \vdots \\ \frac{\partial \boldsymbol{\sigma}_{\mathbf{z}}^{(D)}}{\partial z_1} & \frac{\partial \boldsymbol{\sigma}_{\mathbf{z}}^{(D)}}{\partial z_2} & \cdots & \frac{\partial \boldsymbol{\sigma}_{\mathbf{z}}^{(D)}}{\partial z_d} \end{bmatrix} \tag{33}$$

it is easy to see that $\mathbb{E}_{p(\epsilon)}[\mathbf{B}^{\mathsf{T}}\mathbf{B}] = \left(\mathbf{J}_{\mathbf{z}}^{(\boldsymbol{\sigma})}\right)^{\mathsf{T}} \mathbf{J}_{\mathbf{z}}^{(\boldsymbol{\sigma})}$. So the expectation of the induced Riemannian metric in the latent space by the generator is

$$\bar{\mathbf{M}}_{\mathbf{z}} = \left(\mathbf{J}_{\mathbf{z}}^{(\boldsymbol{\mu})}\right)^{\mathsf{T}} \mathbf{J}_{\mathbf{z}}^{(\boldsymbol{\mu})} + \left(\mathbf{J}_{\mathbf{z}}^{(\boldsymbol{\sigma})}\right)^{\mathsf{T}} \mathbf{J}_{\mathbf{z}}^{(\boldsymbol{\sigma})} \tag{34}$$

which concludes the proof. $\square$

## C  INFLUENCE OF VARIANCE ON THE MARGINAL LIKELIHOOD

We trained a VAE on the digits 0 and 1 of the MNIST scaled to $[-1, 1]$. We randomly split the data to $90\%$ training and $10\%$ test data, ensuring balanced classes. First, we only trained the encoder and the mean function of the decoder. Then, keeping these fixed, we trained two variance functions: one based on standard deep neural network architecture, and the other using our proposed RBF model. Clearly, we have two generators with the same mean function, but different variance functions. Below we present the architectures for the standard neural networks. For the RBF model we used 32 centers and $a = 1$.

| Encoder/Decoder | Layer 1 | Layer 2 | Layer 3 |
|---|---|---|---|
| $\boldsymbol{\mu}_{\phi}$ | 64, (*softplus*) | 32, (*softplus*) | $d$, (*linear*) |
| $\boldsymbol{\sigma}_{\phi}$ | 64, (*softplus*) | 32, (*softplus*) | $d$, (*softplus*) |
| $\boldsymbol{\mu}_{\theta}$ | 32, (*softplus*) | 64, (*softplus*) | $D$, (*tanh*) |
| $\boldsymbol{\sigma}_{\theta}$ | 32, (*softplus*) | 64, (*softplus*) | $D$, (*softplus*) |

The numbers corresponds to the layer size together with the activation function in parenthesis. Further, the mean and the variance functions share the weights of the first layer. The input space dimension is $D = 784$. Then, we computed the marginal likelihood $p(\mathbf{x})$ of the test data using Monte Carlo as:

$$p(\mathbf{x}) = \int_{\mathcal{Z}} p(\mathbf{x}|\mathbf{z})p(\mathbf{z})\mathrm{d}\mathbf{z} \simeq \frac{1}{S} \sum_{s=1} p(\mathbf{x}|\mathbf{z}_s), \quad \mathbf{z}_s \sim p(\mathbf{z}) \tag{35}$$

using $S = 10000$ samples. The generator with the standard variance function achieved -68.25 mean log-marginal likelihood, while our proposed model -50.34, where the higher the better.

The reason why the proposed RBF model performs better can be easily analyzed. The marginal likelihood under the Monte Carlo estimation is, essentially, a large Gaussian mixture model with equal weights $\frac{1}{S}$. Each mixture component is defined by the generator through the likelihood $p(\mathbf{x}|\mathbf{z}) = \mathcal{N}\left(\mathbf{x} \mid \boldsymbol{\mu}_{\theta}(\mathbf{z}), \mathbb{I}_D \boldsymbol{\sigma}_{\theta}^2(\mathbf{z})\right)$. Considering the variance term, the standard neural network approach is trained on the given data points and the corresponding latent codes. Unfortunately, its behavior is arbitrary in regions where there are not any encoded data. On the other hand our proposed model assigns large variance to these regions, while on the regions where we have latent codes its behavior will be approximately the same with the standard neural network. This implies that the resulting marginal likelihood $p(\mathbf{x})$ for the two models are highly similar in regions of high data density, but significantly different elsewhere. The RBF variance model ensures that mixture components in these regions have high variance, whereas the standard architecture assign arbitrary variance. Consequently, the RBF-based $p(\mathbf{x})$ assigns minimal density to regions with no data, and, thus, attains higher marginal likelihood elsewhere.

# D  IMPLEMENTATION DETAILS FOR THE EXPERIMENTS

---
**Algorithm 1** The training of a VAE that ensures geometry
---
**Output:** the estimated parameters of the neural networks $\theta, \phi, \psi$
  1: Train the $\boldsymbol{\mu}_\phi, \boldsymbol{\sigma}_\phi, \boldsymbol{\mu}_\theta$ as in Kingma & Welling (2014), keeping $\boldsymbol{\sigma}_\psi$ fixed.
  2: Train the $\boldsymbol{\sigma}_\psi$ as explained in Sec. 4.1.
---

**Details for Experiments 5.1, 5.2 & 5.3.**   The pixel values of the images are scaled to the interval $[0, 1]$. We use for the functions $\boldsymbol{\mu}_\phi, \boldsymbol{\sigma}_\phi, \boldsymbol{\mu}_\theta$ multilayer perceptron (MLP) deep neural networks, and for the $\boldsymbol{\beta}_\psi$ the proposed RBF model with 64 centers, so $\mathbf{W} \in \mathbb{R}^{D \times 64}$ and the parameter $a$ of Eq. 11 is set to 2. We used $L_2$ regularization with parameter equal to $1e^{-5}$.

| Encoder/Decoder | Layer 1 | Layer 2 | Layer 3 |
|---|---|---|---|
| $\boldsymbol{\mu}_\phi$ | 64, (*tanh*) | 32, (*tanh*) | $d$, (*linear*) |
| $\boldsymbol{\sigma}_\phi$ | 64, (*tanh*) | 32, (*tanh*) | $d$, (*softplus*) |
| $\boldsymbol{\mu}_\theta$ | 32, (*tanh*) | 64, (*tanh*) | $D$, (*sigmoid*) |

The number corresponds to the size of the layer, and in the parenthesis the activation function. For the encoder, the mean and the variance functions share the weights of the Layer 1. The input space dimension $D = 784$. After the training, the geodesics can be computed by solving Eq. 7 numerically. The LAND mixture model is fitted as explained in (Arvanitidis et al., 2016).

**Details for Experiments 5.4.**   In this experiment we used Convolutional Variational Auto-Encoders. The pixel values of the images are scaled to the interval $[0, 1]$. For the $\boldsymbol{\beta}_\psi$ we used the proposed RBF model with 64 centers and the parameter $a$ of Eq. 11 is set to 2.

Considering the variance network during the decoding stage, the RBF generates an image, which represents intuitively the total variance of each pixel for the decoded final image, but in an initial sub-sampled version. Afterwards, this image is passed through a sequence of deconvolution layers, and at the end will represent the variance of every pixel for each RGB channel. However, it is critical that the weights of the filters must be clipped during the training to $\mathbb{R}_+$ to ensure positive variance.

| Encoder | Layer 1 (Conv) | Layer 2 (Conv) | Layer 3 (MLP) | Layer 4 (MLP) |
|---|---|---|---|---|
| $\boldsymbol{\mu}_\phi$ | 32, 3, 2, (*tanh*) | 32, 3, 2, (*tanh*) | 1024, (*tanh*) | $d$, (*linear*) |
| $\boldsymbol{\sigma}_\phi$ | 32, 3, 2, (*tanh*) | 32, 3, 2, (*tanh*) | 1024, (*tanh*) | $d$, (*softplus*) |

For the convolutional and deconvolutional layers, the first number is the number of applied filters, the second is the kernel size, and third is the stride. Also, for the encoder, the mean and the variance functions share the convolutional layers. We used $L_2$ regularization with parameter equal to $1e^{-5}$.

| Decoder | L. 1 (MLP) | L. 2 (MLP) | L. 3 (DE) | L. 4 (DE) | L. 5 (DE) | L.6 (CO) |
|---|---|---|---|---|---|---|
| $\boldsymbol{\mu}_\theta$ | 1024, $(t)$ | $D/4$, $(t)$ | 32, 3, 2, $(t)$ | 32, 3, 2, $(t)$ | 3, 3, 1, $(t)$ | 3, 3, 1, $(s)$ |

For the decoder, the acronyms (DE) = Deconvolution, (CO) = Convolution and $(t)$, $(s)$ stand for *tanh* and *sigmoid*, respectively. Also, $D = width \times height \times channels$ of the images, in our case 64,64,3. For all the convolutions and deconvolutions, the padding is set to *same*. We used $L_2$ regularization with parameter equal to $1e^{-5}$.

| Decoder | Layer 1 (RBF) | Layer 2 (Deconv) | Layer 3 (Conv) |
|---|---|---|---|
| $\boldsymbol{\beta}_\psi$ | $W \in \mathbb{R}^{(D/2) \times 64}$ | 1, 3, 2 (*linear*) | 3, 3, 1 (*linear*) |

The Brownian motion over the Riemannian manifold in the latent space is presented in Alg. 2.

---

**Algorithm 2** Brownian motion on a Riemannian manifold

---

**Input:** the starting point $\mathbf{z} \in \mathbb{R}^{d \times 1}$, stepsize $s$, number of steps $N_s$, the metric tensor $\mathbf{M}(\cdot)$.
**Output:** the random steps $\mathbf{Z} \in \mathbb{R}^{N_s \times d}$.
1: **for** $n = 0$ to $N_s$ **do**
2:    $\mathbf{L}, \mathbf{U} = eig\left(\mathbf{M}(\mathbf{z})\right),$      ($\mathbf{L}$: eigenvalues, $\mathbf{U}$: eigenvectors)
3:    $\mathbf{v} = \mathbf{U}\mathbf{L}^{-\frac{1}{2}}\boldsymbol{\epsilon},$     $\boldsymbol{\epsilon} \sim \mathcal{N}(\mathbf{0}, \mathbb{I}_d)$
4:    $\mathbf{z} = \mathbf{z} + s \cdot \mathbf{v}$
5:    $\mathbf{Z}(n,\ :) = \mathbf{z}$
6: **end for**

---

