# OpenReview forum: "Latent Space Oddity: on the Curvature of Deep Generative Models"
_ICLR.cc/2018/Conference — Accept (Poster)_

### Official Review · AnonReviewer2 · 2017-11-23
**Decent outline of standard Riemannian geometry in generative models but no novelty**

**Rating:** 3
**Confidence:** 4

**Review:**

The paper investigates the geometry of deep generative models. In particular, it describes the geometry of the latent space when giving it the (stochastic) Riemannian geometry inherited from the embedding in the input space described by the generator function. The authors describe the geometric setting, how distances in the latent space can be interpreted with the non-Euclidean geometry, and how interpolation, probability distributions and random walks can be constructed.

While the paper makes a decent presentation of the geometry of the generative setting, it is not novel. It is well known that (under certain conditions) the mapping described by the generator function is a submanifold of the input space. The latent space geometry is nothing but the submanifold geometry the image f(Z) inherits from the Euclidean geometry of X. Here f is the generator mapping f:Z->X. The latent space distances and geodesics corresponds to distances and geodesics on the submanifold f(Z) of X. Except for f being stochastic, the geometry is completely standard. It is not surprising that distances inherited from X are natural since they correspond to the Euclidean length of minimal curves in the input space (and thus the data representation) when restricting to f(Z).

I cannot identify a clear contribution or novelty in the paper which is the basis for my recommendation of rejection of the paper.

---

> ### Author Response · Authors · 2017-12-05
> **Reply**
>
> "I cannot identify a clear contribution or novelty in the paper which is the basis for my recommendation of rejection of the paper."
>
> It is true that if f:Z->X is a sufficiently well-behaved deterministic function then it is completely standard differential geometric analysis to show that it generates a submanifold of R^D. This is, however, not the point of the present paper. For VAEs (and related models), the generator function f is stochastic and then standard geometric analysis no longer applies (now we get random submanifolds; a topic that is not commonly discussed). A brief summary of our contributions are then:
>
> *) We show how the expected metric of the stochastic generator has a particularly appealing form, where the Jacobian of the generator mean captures the shape of the submanifold, while the Jacobian of the generator variance pushes geodesics towards regions of high data density.
>
> *) We show that standard estimators of generator variance are rather arbitrary and do not make much sense. We provide a simple approach for improving this that works quite well in our settings, such the learned geometry becomes useful.
>
> *) We show that in a deep learning context  Riemannian metrics are quite useful for getting a better understanding of the latent space. Furthermore, we demonstrate that knowledge of the geometry improves results on many tasks where latent variable models are commonly used today.
>
> We would argue that these are all substantial contributions. From the reviewer comments it would appear that our insights were already known; if this is indeed the case, we would appreciate specific pointers to the literature where the stochastic Riemannian metric of deep generative models are discussed. We are not aware of any such previous publications.

---

### Official Review · AnonReviewer1 · 2017-11-24
**good paper**

**Rating:** 7
**Confidence:** 4

**Review:**

In the paper the authors analyse the latent space generated by the variational autoencoder (VAE). They show that this latent space is imbued by a Riemannian metric and that this metric can be easily computed in terms of mean and variance functions of the corresponding VAE. They also argue that the current variance estimates are poor in regions without data and propose a meaningful variance function instead. In the experiments section the authors evaluate the quality and meaningfulness of the induced Riemannian metric.

There are minor grammatical errors and the paper would benefit from proofreading.

In the introduction the authors argue that points from different classes being close to each other is a misinterpretation of the latent space. An argument against would be that such a visualisation is simply bad. A better visualisation would explain the data structure without the need for an additional visualisation of the metric of the latent space.

In section 2, the multiplication symbol (circle with dot inside) is not defined.

It is not clear from the paper what the purpose of eq. 7 is, as well as most of the section 3. Only in appendix C, it is mentioned that eq. 7 is solved numerically to compute Riemannian distances, though it is still not clear how exactly this is achieved. I think this point should be emphasized and clarified in section 3.

In section 4, it says proof for theorem 1 is in appendix B. Appendix B says it proves theorem 2. Unfortunately, it is not clear how good the approximation in eq. 9 is.

Is theorem 1 an original result by the authors? Please emphasize.

In Fig. 6, why was 7-NN used, instead of k-means, to colour the background?

I think that the result from the theorem 1 is very important, since the estimation of the Riemannian metric is usually very slow. In this regard, it would be very interesting to know what the total computational complexity of the proposed approach is.

---

> ### Author Response · Authors · 2017-12-05
> **Replies**
>
> "...An argument against would be that such a visualisation is simply bad...."
>
> We agree that this visualization is bad, yet it is ever-present in the literature as it reflect the Euclidean structure that is almost-always imposed on the latent space (e.g. it is common to add and subtract latent vectors for style transfer). We use this visualization merely to show that the Euclidean assumption need not be particularly good.
>
> As a side-remark, we find that having the volume measure of the metric as a background color (as done in Figs. 3, 5, 7, 8 and 10) helps quite a bit as a simple visualization tool. Chris Bishop's "magnification factor" for visualizing the GTM model is identical (except the GTM has constant generator-variance), so there is some experience in the community already with such a visualization.
>
> C. M. Bishop, M. Svensén, and C. K. I. Williams. Magnification factors for the SOM and GTM algorithms. In Proceedings 1997 Workshop on Self-Organizing Maps, Helsinki University of Technology, Finland., pages 333-338, 1997.
>
>
> "In section 4, it says proof for theorem 1 is in appendix B. Appendix B says it proves theorem 2."
>
> We have fixed the incorrect reference in the appendix; it should indeed state that this is a proof of theorem 1.
>
> "Unfortunately, it is not clear how good the approximation in eq. 9 is."
>
> The variance of the metric drops as O(1/D); in the limit D->infinity the variance vanishes. We find that for high-dimensional data (e.g. images) the variance is effectively zero, and the approximation is quite good. If the data space is low-dimensional, we expect that the approximation is less well-behaved.
>
> "Is theorem 1 an original result by the authors?"
>
> Yes, theorem 1 is a novel contribution. However, the derivation of the result is purely mechanical, and we only state it as a theorem to
> a) emphasize the result (it is important for the paper), and
> b) make it easier to push its derivation to an appendix.
>
> "In Fig. 6, why was 7-NN used, instead of k-means, to colour the background?"
>
> We wanted the background color of the figure to resemble a "ground truth", so it seemed more natural to use a more sensitive model than k-means for this. We're happy to change this, if that is deemed better.
>
> "it would be very interesting to know what the total computational complexity of the proposed approach is."
>
> The complexity of computing the metric merely amounts to computing Jacobians of the generator; the complexity of this operation depends very-much on the network architecture of the generator. In practice, it can be time consuming as TensorFlow (which we used) does not support Jacobians (only gradients), so theoretical computational complexity does not reflect the runtime of current tools.
>
>
> "There are minor grammatical errors and the paper would benefit from proofreading."
>
> We will send the paper to an external agency for proof-reading. This usually takes a few days, after which we will update the paper.
>
> "In section 2, the multiplication symbol (circle with dot inside) is not defined."
>
> Good catch; we have fixed this (it is the element-wise product).
>
> "It is not clear from the paper what the purpose of eq. 7 is, as well as most of the section 3. Only in appendix C, it is mentioned that eq. 7 is solved numerically to compute Riemannian distances, though it is still not clear how exactly this is achieved."
>
> Fair point. This material is mostly included for completeness as it is otherwise not possible to build a numerical implementation of the proposed models. We solve Eq. 7 numerically using Matlab's 'bvp5c', so the equation translates directly into an implementation of an algorithm for computing geodesics. We have made some changes to the paper to reflect this.

---

### Official Review · AnonReviewer3 · 2017-11-28
**Interesting view on the induced Riemannian geometry of latent space in deep generative models. Several important applications are explored, but not in depth**

**Rating:** 7
**Confidence:** 3

**Review:**

The paper makes an important observation: the generating function of a generative model (deep or not) induces a (stochastic) Riemannian metric tensor on the latent space. This metric might be the correct way to measure distances in the latent space, as opposed to the Euclidean distance.

While this seems obvious, I had actually always thought of the latent space as "unfolding" the data manifold as it exists in the output space. The authors propose a different view which is intriguing; however, they do not, to the best of my understand, give a definitive theoretical reason why the induced Riemannian metric is the correct choice over the Euclidean metric.

The paper correctly identifies an important problem with the way most deep generative models evaluate variance. However the solution proposed seems ad-hoc and not particularly related to the other parts of the paper. While the proposed variance estimation (using RBF networks) might work in some cases, I would love to see (perhaps in future work) a much more rigorous treatment of the subject.

Pros:
1. Interesting observation and mathematical development of a Riemannian metric on the latent space.

2. Good observation about the different roles of the mean and the variance in determining the geodesics: they tend to avoid areas of high variance.

3. Intriguing experiments and a good effort at visualizing and explaining them. I especially appreciate the interpolation and random walk experiments. These are hard to evaluate objectively, but the results to hint at the phenomena the authors describe when comparing Euclidean to Riemannian metrics in the latent space.

Cons:
1. The part of the paper proposing new variance estimators is ad-hoc and is not experimented with rigorously, comparing it to other methods in terms of calibration for example.

Specific comments:
1. To the best of my understanding eq. (2) does not imply that the natural distance in Z is locally adaptive. I think of eq (2) as *defining* a type of distance on Z, that may or may not be natural. One could equally argue that the Euclidean distance on z is natural, and that this distance is then pushed forward by f to some induced distance over X.

2. In the definition of paths \gamma, shouldn't they be parametrized by arc-length (also known as unit-speed)? How should we think of the curve \gamma(t^2) for example?

3. In Theorem 2, is the term "input dimension" appropriate? Perhaps "data dimension" is better?

4. I did not fully understand the role of the LAND model. Is this a model fit AFTER fitting the generative model, and is used to cluster Z like a GMM ? I would appreciate a clarification about the context of this model.

---

> ### Author Response · Authors · 2017-12-05
> **Replies**
>
> === Which metric is "correct" ===
>
> It is not unreasonable to view the latent space as an "unfolding" of the data manifold. However, making this unfolding is generally not possible without squeezing and stretching, which introduce significant curvature in the latent space. From that point of view, the "mean term" of our proposed metric, captures the squeezing and stretching, while the "variance term" captures the inherent uncertainty of the latent space, which appear in regions of low data density (i.e. where the "unfolding" is of poor quality due to lack of data). So, our work does not conflict with an "unfolding" interpretation of the latent space; rather it reflect the squeezing, stretching and uncertainty needed to unfold.
>
> The stochastic Riemannian metric is the "correct choice" when infinitesimal distances along the data manifold are meaningful in the input space. This is a valid assumption for many data sources, but not necessarily for all. Two remarks for when the assumption is not valid:
>
>   1) Our proposed metric may still be useful as the variance term force shortest paths to go near regions of high data density; this is most-often an useful property.
>
>   2) When the Euclidean distance is not useful infinitesimally in input space, it may be possible to pick another inner product in the input space which is sensible infinitesimally. Our proposed formalism can then be used to pull back this inner product to the latent space.
>
> In practice, we do not work with the stochastic Riemannian metric, but rather with its expectation. This simplify both mathematics and computations, and can be justified as the distribution of the metric concentrates for high-dimensional data. It is, nonetheless, an approximation.
>
> === The variance network ===
>
> We largely agree that the proposed variance estimator is ad hoc. We tried several architectures of the variance network before converging on this rather simple RBF network, which is, so far, the only one that worked reliably well for us.
>
> It is worth noting that in standard architectures the variance network is only trained in regions where there is data. Consequently, feedforward networks with standard sigmoid-like activations will extrapolate the variance in a somewhat arbitrary manor. We tried several variants of variance networks (both with and without weight-sharing with the mean network), and all had low-quality extrapolations. This resulted in geodesics that did not follow the trend of the data, which defeated our purposes. The images of "standard" variance networks in the paper (Figs. 4 and 5) are indeed only anecdotal examples of how standard feedforward networks fare, but they are good representatives of our experiences.
>
> We're happy to add additional examples of the geometry induced by standard variance networks if so requested. We did not include those as they work quite poorly (practically useless).
>
>
> === Specific comments ===
> "...eq. (2) does not imply that the natural distance in Z is locally adaptive..."
>
> We agree, and have changed the wording after Eq. 2. We simply meant that the distance changes locally; from Eq. 2 it is indeed not evident that the distance adapt to the data. However, if the generator variance is small in regions of high data density and large otherwise, then Theorem 1 provide a strong hint that the distance measure will indeed locally adapt.
>
> "One could equally argue that the Euclidean distance on z is natural, and that this distance is then pushed forward by f to some induced distance over X."
>
> In VAEs the latent space is only optimized to ensure that the latent variables approximately follow a unit Gaussian, so here we do not see a particular strong argument for using the Euclidean distance over Z. That being said, we are not trying to argue that our proposed metric is always the best one, merely that it is a natural choice with some very appealing properties.
>
> "In the definition of paths \gamma, shouldn't they be parametrized by arc-length (also known as unit-speed)?"
>
> Computationally, geodesics are found by solving a boundary value problem (Eq. 7) numerically. Here we need to specify a start- and end-time, which we arbitrarily choose as t=0 and t=1, respectively. Then the solution curve is approximately constant-speed, but not unit speed (its speed is scaled by the length of the geodesic).
>
> "In Theorem 2, is the term "input dimension" appropriate? Perhaps "data dimension" is better?"
>
> We agree and have changed this.
>
> "I did not fully understand the role of the LAND model. Is this a model fit AFTER fitting the generative model..."
>
> Yes, the LAND is fitted post hoc. This is true for all experiments: we first fit a VAE and then analyze the latent variables according to the implied geometry.

---

### Author Response · Authors · 2017-12-05
**Thank you**

We thank all reviewers for their comments. We will reply to these comments separately.

---

### Author Response · Authors · 2017-12-21
**Paper updated**

We have updated the paper with the following two changes:

1) As promised to AnonReviewer1 the paper has been professionally proof-read.

2) AnonReviewer3 asked about the proposed variance function based on RBF networks. We have added an extra experiment to Appendix C which demonstrate that our proposed model (besides improving the latent geometry) improves marginal likelihood of held-out data. While we proposed this variance function to get a well-behaved geometry, this experiment shows that it also generally improves density modeling compared to standard models.

---

### Decision · Program_Chairs · 2018-01-29
**ICLR 2018 Conference Acceptance Decision**

**Decision:**

Accept (Poster)

**Comment:**

This paper characterizes the induced geometry of the latent space of deep generative models. The motivation is established well, such that the paper convincingly discusses the usefulness derived from these insights. For example, the results uncover issues with the currently used methods for variance estimation in deep generative models. The technique invoked to mitigate this issue does feel somehow ad-hoc, but at least it is well motivated.

One of the reviewers correctly pointed out that there is limited novelty in the theoretical/methodological aspect. However, I agree with the authors’ rebuttal in that characterizing geometries on stochastic manifolds is much less studied and demonstrated, especially in the deep learning community. Therefore, I believe that this paper will be found useful by readers of the ICLR community, and will stimulate future research.